# Seasonal Dynamics of Heavy Metal Concentrations in Water and Fish from Hakaluki *Haor* of Bangladesh

**Abu Shahadat Mohammad Sadequr Rahman Bhuyain [1], Sanzib Kumar Barman [2][ID], Md. Motaher Hossain [1], Mohammad Mehedi Hasan Khan [3][ID], Khadizatul Kubra Mim [4] and Sabuj Kanti Mazumder [5],*[ID]**

[1] Department of Fisheries Technology and Quality Control, Faculty of Fisheries, Sylhet Agricultural University, Sylhet 3100, Bangladesh; raselsau37@gmail.com (A.S.M.S.R.B.); motaher.ftqc@sau.ac.bd (M.M.H.)
[2] Department of Fishery Resources Conservation and Management, Faculty of Fisheries and Ocean Sciences, Khulna Agricultural University, Khulna 9100, Bangladesh; sanzibarm.sau@gmail.com
[3] Department of Biochemistry and Chemistry, Faculty of Biotechnology and Genetic Engineering, Sylhet Agricultural University, Sylhet 3100, Bangladesh; khanmmh.biochem@sau.ac.bd
[4] Department of Aquatic Resource Management, Faculty of Fisheries, Sylhet Agricultural University, Sylhet 3100, Bangladesh; mimosau11@gmail.com
[5] Department of Genetics and Fish Breeding, Bangabandhu Sheikh Mujibur Rahman Agricultural University, Gazipur 1706, Bangladesh
* Correspondence: sabujgfb@bsmrau.edu.bd

**Abstract:** Food safety is currently a serious concern due to the health risks associated with food intake, particularly due to heavy metal contamination. Therefore, the present study was conducted to investigate the heavy metals concentration in water and fishes collected from Hakaluki *haor*, Bangladesh. Three important fish species, *Labeo rohita, Cirrhinus cirrhosis*, and *Labeo calbasu*, together with water samples, were analyzed for heavy metals (Pb, Cr, and Cd), respectively. Considering four seasons, namely the monsoon, post-monsoon, winter, and pre-monsoon, a total of 72 fish samples were collected from three fishing stations from June 2017 to May 2018. The results showed that the total mean concentration of metals in water (mg L$^{-1}$) was found to be in the order of Pb (0.125 ± 0.058) > Cr (0.026 ± 0.012) > Cd (0.001 ± 0.0002) within the maximum permissible limits set by the EU and WHO, except for Pb concentrations. Similarly, the total mean concentrations of Pb in fish (µg g$^{-1}$) were found in order of *L. rohita* (0.388 ± 0.291) > *Cirrhinus cirrhosus* (0.334 ± 0.236) > *L. calbasu* (0.251 ± 0.117) greater than the maximum permissible limits (0.3 µg g$^{-1}$) set by FIQC, except for *L. calbasu*. However, the mean concentrations of Cr and Cd in fish were found to be below the maximum permissible limits of FAO and FIQC, respectively. The quantity of heavy metal contamination in this *haor* indicates that the situation is worrying for the region's biota and residents. However, to protect public health and reduce environmental risk, the appropriate authorities should oversee and monitor it with strong hands.

**Keywords:** seasonal dynamics; heavy metals; Hakaluki *haor*; Bangladesh

## 1. Introduction

A *haor* is a floodplain depression that has a bowl-like form and particular hydro-ecological characteristics. It is made up of a patchwork of wetlands, which includes rivers, streams, irrigation canals, and a substantial area of agricultural land that is sporadically inundated [1]. The largest freshwater inland wetland environment in South Asia and in Bangladesh is called Hakaluki *haor* [2,3]. It is one of Bangladesh's four main "mother fisheries" and serves as a crucial habitat and breeding ground for fish and other aquatic species [4]. This *haor* is home to more than 100 different fish species, among one-third of them being endangered [3,5]. Around 200,000 people reside in the *haor*'s vicinity, all of whom are reliant on its resources in some way [5].

Heavy metals are major non-biodegradable environmental pollutants that can cause cytotoxic, mutagenic, and carcinogenic effects in animals [6]. At present, heavy metal

pollution has become a serious environmental concern [7] as aquatic organisms, especially fish, have the ability to accumulate heavy metals from various sources, including sediments, soil erosion and runoff, air depositions of dust and aerosol, and discharges of wastewater [8,9]. The primary source of heavy metals in the environment, however, has been determined to be the mining industry, with additional sources including industrial and domestic waste discharges, traffic-related pollutants such as vehicle exhaust, brake linings, tyre wear, asphalt wear, and gasoline and oil leakage [10]. Fish may concentrate enormous amounts of metals from the water, and they are frequently at the top of the aquatic food chain [11]. Fish bodies can absorb heavy metals in three different ways: through the skin, gills, and digestive system [12]; they then build up in several organs, including the liver, kidneys, spleen, and gonads [13]. Thus, the accumulation of these metals in aquatic creatures can have a long-term impact on biogeochemical cycling in the ecosystem [14].

Therefore, the food safety concept has become a big worry in recent years as the eating of fish has grown in popularity among health-conscious people due to its high protein content, low saturated fat content, and omega fatty acid concentration, all of which are known to promote excellent health [15]. As a result, the growing demand for food safety has spurred studies to consider the risk of consuming heavy metal-contaminated food [16]. In the *haor* regions of Bangladesh, where fish and water had a low to extraordinarily high bio-accumulation factor, a study was undertaken to determine the amounts of heavy metals in fish, sediment, and water. Additionally, it was discovered that the level of heavy metal contamination in the *haor* region is worrying for both the local biota and people who live there [17]. However, Hakaluki *haor* has a strong potential for fisheries production, and the local people in this area eat fish caught in this *haor* and rely on it in various ways. Moreover, the wetlands, especially the *haor*, plays a major role in the economy of the Sylhet region of Bangladesh due to its geographical position [18,19]. Thus, assessing heavy metal contamination in fish and water bodies has attained global attention. Hence, the current investigation was conducted for the first time on the Hakaluki *haor*, with the specific goal of determining the concentrations of heavy metals such as Pb, Cr, and Cd in the water, as well as three commonly available and popular Indian Major Carp fish species: Rui (*Labeo rohita*), Mrigal (*Cirrhinus cirrhosus*), and Kalibaus (*Labeo calbasu*), respectively.

Because fish muscles are the most popular fish dish in Bangladesh, the current study focused on the levels of heavy metals in them. Additionally, it has been proven that some fish living in contaminated areas may accumulate significant amounts of metals in their tissues, sometimes exceeding the upper allowed limits [20]. The selected fishes of the present study are mostly available and commonly consumed by the local people of Bangladesh. On the other hand, although a few studies have been conducted on Hakaluki *haor* [17,21], no research has so far been conducted on the heavy metal accumulation in the water and fishes of this *haor*. For this reason, this study has significant importance for identifying the safety measures for the aforementioned consumed fishes for local consumption as well as for export.

## 2. Materials and Methods

### 2.1. Study Area

Three fishing sites in Hakaluki *haor* were selected for sample collection (Figure 1), namely, Gilachhara of Fenchuganj Upazila (site-1), the Islamganj Bazar area of Kulaura Upazila (site-2), and Kanungo Bazar bazaar of Barlekha Upazila (site-3), respectively. Fishing activity and the distance between the industry's location and the water body were taken into consideration while choosing the sampling locations. However, fishing time and the availability of fish species were also taken into consideration around the year.

### 2.2. Sample Collection and Preservation

A total of seventy-two (72) fish samples, of which there were 24 samples for each individual, were collected and analyzed on a seasonal basis from June 2017 to May 2018. In

the meantime, samples were also collected from each sampling site using a 50 mL centrifuge tube. Moreover, the fish samples were collected individually in separate polythene bags and kept isolated inside an ice box to ensure better quality before being brought to the Fisheries Technology and Quality Control (FTQC) laboratory of Sylhet Agricultural University, Sylhet, for further morphometric measurements (Table 1). After that, the fish samples were double bagged in new, separate plastic bags and sealed and labeled accordingly, and finally transferred to the Toxicology Laboratory of the Institute of Food Science and Technology (IFST), Bangladesh Council of Scientific and Industrial Research (BCSIR), Dhaka, Bangladesh, where the heavy metal analysis was performed. However, the packed samples were kept in the laboratory refrigerator at $-20\ ^\circ$C until the chemical analysis.

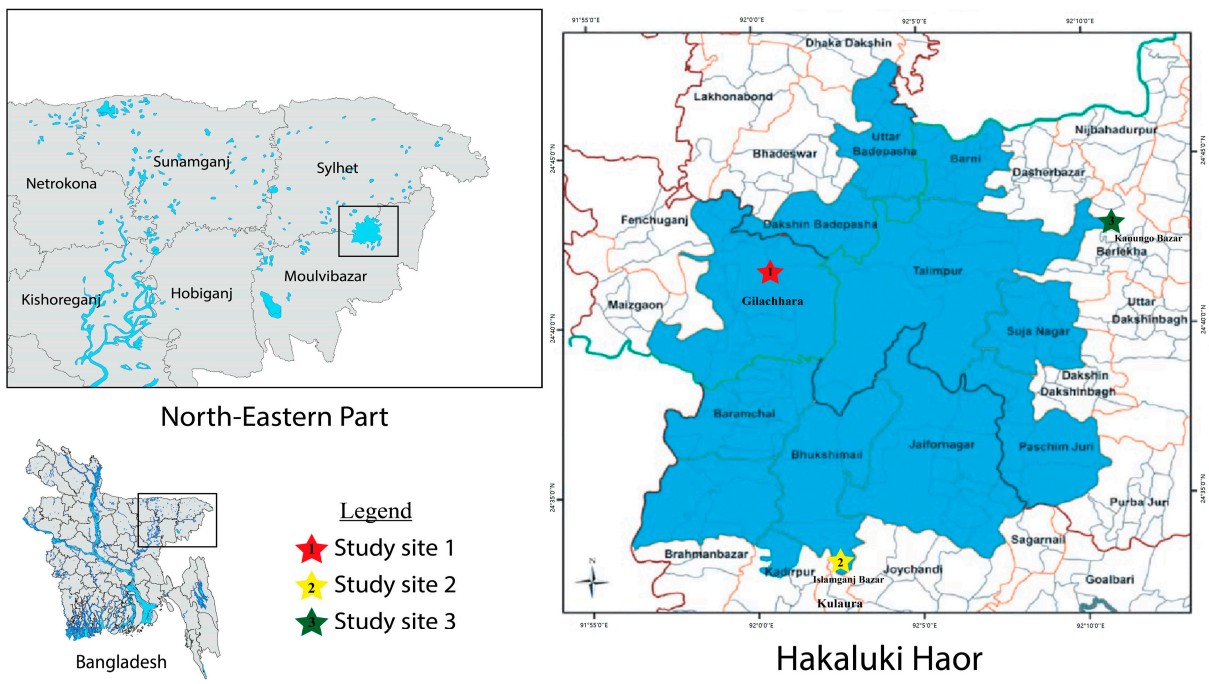

**Figure 1.** Map showing three sampling stations in Hakaluki *haor*.

**Table 1.** Morphometric characteristics of the collected fish samples clarified by species and area (Values are presented as mean $\pm$ SD).

| Species | Sites | Sample Number | Weight (g) | Length (cm) | Age (Years) | Sex Ratio (m/f) |
|---|---|---|---|---|---|---|
| *Labeo rohita* | Site-1 | 8 | 343.5 $\pm$ 102.12 | 34.89 $\pm$ 3.66 | 1.23 $\pm$ 0.23 | 13/8 |
| | Site-2 | 8 | 377.15 $\pm$ 92.1 | 36.9 $\pm$ 5.74 | 1.38 $\pm$ 0.28 | |
| | Site-3 | 8 | 365.19 $\pm$ 118.58 | 37.29 $\pm$ 8.07 | 1.4 $\pm$ 0.35 | |
| *Cirrhinus cirrhohus* | Site-1 | 8 | 156.65 $\pm$ 32.63 | 27.73 $\pm$ 2.65 | 0.94 $\pm$ 0.16 | 9/12 |
| | Site-2 | 8 | 216.99 $\pm$ 46.26 | 33.19 $\pm$ 6.76 | 1.4 $\pm$ 0.25 | |
| | Site-3 | 8 | 195.11 $\pm$ 50.55 | 33 $\pm$ 5.59 | 1.19 $\pm$ 0.22 | |
| *Labeo calbasu* | Site-1 | 8 | 51.28 $\pm$ 6.96 | 20.83 $\pm$ 3.71 | 0.7 $\pm$ 1.22 | 11/10 |
| | Site-2 | 8 | 51.07 $\pm$ 9.25 | 21.03 $\pm$ 2.87 | 0.75 $\pm$ 0.17 | |
| | Site-3 | 8 | 59.15 $\pm$ 10.44 | 21.84 $\pm$ 4.82 | 0.88 $\pm$ 0.19 | |

### 2.3. Sample Preparation

The water and the fish samples were taken from the refrigerator, and the fish samples were thawed at room temperature. The fish samples were prepared by removing the scales, fins, and viscera and then washed with clean water. After that, the required amount of fish muscle (edible part) was finely chopped and blended manually on a chopping board. Then, the chopped samples were gathered into a clear plastic bag with appropriate labelling.

### 2.4. Sample Analysis and Heavy Metals Determination

An Atomic Absorption Spectrophotometer (AAS) (Model: AA-6300, Shimadzu, Kyoto, Japan) was used to perform the routine analytical method on the collected water and fish samples (Table 2).

**Table 2.** Spectral lines used in emission measurements and the instrumental detection limit for the elements tested by using AAS.

| Elements | Wavelength (nm) | Instrumental Detection Limit (mg L$^{-1}$) |
|:---:|:---:|:---:|
| Pb | 217.0 | 0.004 |
| Cr | 357.9 | 0.001 |
| Cd | 228.8 | 0.001 |

Each specimen's homogenized muscles weighed 5 g, and 10 mL of water samples were taken for the heavy metal extraction process. After placing the sample inside a fume hood and burning it on a hot plate, 2 mL (4/5 drops) of concentrated nitric acid ($HNO_3$) was added to it. The samples were subsequently maintained in the furnace, where they were burned for 5–6 hrs at 600 °C, turning them into ash. To begin digesting the samples inside the fume hood, 10 mL of HCL (5 M) was added at that time together with the ash. The samples were boiled until they were colorless or transparent and then the extracts were filtered through a 12.5 cm Whatman No. 1 filter paper into a 100 mL volumetric flask within the fume hood before filling it to the top with de-ionized water. Finally, multiple wavelengths of the Atomic Absorption Spectrophotometer (AAS) were used to assess the heavy metals. Air served as the oxidizer, while argon gas served as the fuel in an AAS. Windows XP/2000 was required for the "WizAArd" program that runs in the AAS. The digested samples were sucked into an air acetylene flame that was rich in fuel, and the metal concentrations were calculated using calibration curves made from standard solutions. The average results of three replicate samples were used for each determination, which validated the previous studies [22,23].

### 2.5. Blank Preparation

To guarantee that the samples and chemicals employed were not contaminated, acid blanks (laboratory blanks) were prepared at each step of the digestion process of the samples. They lacked a fish sample but included the same digestive reagents and acid ratios as the original samples. After digestion, acid blanks were handled as samples and diluted with the same agent. In order to be sure that the equipment was reading just the precise amounts of heavy metals in genuine samples, they were tested with an atomic absorption spectrophotometer before testing the real samples, and their readings were subtracted. An acid blank was used to adjust each pair of the digested samples.

### 2.6. Physical and Chemical Parameters of Water

Water samples were taken on-site using water samplers in order to identify the physicochemical parameters. The temperature and pH were measured using a microprocessor pH meter (Model No. HI 98139, HANNA Test kits, Hanna Instruments Ltd., Vöhringen, Germany). The use of these kits allowed for the analysis of additional parameters, including hardness (mg L$^{-1}$), dissolved oxygen (mg L$^{-1}$) and ammonia (mg L$^{-1}$) [23].

### 2.7. Statistical Analysis

The statistical analyses were carried out using Minitab 17 and Origin 8 software. The one-way analysis of variance (ANOVA) and *t*-test were used to assess whether the metal concentration varied significantly among fish and water samples at different seasons. All of the data were evaluated for normality and homogeneity of variances before being analyzed. At the 0.05 threshold, the data in distinct letters were significant.

## 3. Results

### 3.1. Heavy Metals Concentration in Water

The heavy metal concentrations (mg L$^{-1}$) in the water samples from the Hakaluki *haor* were significantly different from each of the study sites ($p < 0.05$), which are shown in Table 3, respectively. The results from the current investigation also showed that the mean concentrations (0.125 ± 0.058 mg L$^{-1}$) of Pb were significantly higher in the water sample compared to the others ($p < 0.05$).

**Table 3.** Heavy metal concentrations (mg L$^{-1}$) in water sample of Hakaluki *haor* at different study sites (Data presented as mean ± SD).

| Parameters | Sampling Sites | Seasons | | | | Total Mean ± SD |
|---|---|---|---|---|---|---|
| | | **Monsoon** | **Post-Monsoon** | **Winter** | **Pre-Monsoon** | |
| Pb | Site-1 | 0.242 ± 0.034 [a] | 0.114 ± 0.033 [ab] | 0.088 ± 0.005 [b] | 0.155 ± 0.018 [ab] | |
| | Site-2 | 0.175 ± 0.038 [a] | 0.121 ± 0.051 [a] | 0.086 ± 0.011 [a] | 0.088 ± 0.009 [a] | 0.125 ± 0.058 |
| | Site-3 | 0.175 ± 0.047 [a] | 0.114 ± 0.037 [a] | 0.054 ± 0.015 [a] | 0.092 ± 0.005 [a] | |
| | Mean ± SD | 0.197 ± 0.031 | 0.116 ± 0.003 | 0.076 ± 0.016 | 0.111 ± 0.031 | |
| Cr | Site-1 | 0.083 ± 0.040 [a] | 0.020 ± 0.003 [b] | 0.013 ± 0.001 [b] | 0.011 ± 0.002 [b] | |
| | Site-2 | 0.034 ± 0.008 [a] | 0.028 ± 0.011 [a] | 0.030 ± 0.029 [a] | 0.008 ± 0.001 [a] | 0.026 ± 0.012 |
| | Site-3 | 0.059 ± 0.032 [a] | 0.016 ± 0.002 [b] | 0.007 ± 0.001 [b] | 0.007 ± 0.001 [b] | |
| | Mean ± SD | 0.059 ± 0.02 | 0.021 ± 0.005 | 0.017 ± 0.009 | 0.009 ± 0.002 | |
| Cd | Site-1 | 0.00 0 ± 0.000 [a] | 0.000 ± 0.000 [a] | 0.002 ± 0.001 [a] | 0.000 ± 0.000 [a] | |
| | Site-2 | 0.001 ± 0.0001 [ab] | 0.000 ± 0.000 [b] | 0.005 ± 0.001 [a] | 0.002 ± 0.001 [ab] | |
| | Site-3 | 0.000 ± 0.000 | 0.000 ± 0.000 | 0.000 ± 0.000 | 0.000 ± 0.000 | 0.001 ± 0.0002 |
| | Mean ± SD | 0.0003 ± 0.00004 | 0.000 ± 0.000 | 0.002 ± 0.001 | 0.001 ± 0.0004 | |

Values in a row is the same superscript are not significantly different ($p > 0.05$). Absolute zero values represented below the detection limit.

However, the highest average concentrations of Pb (0.197 ± 0.031 mg L$^{-1}$) and the lowest average levels (0.076 ± 0.016 mg L$^{-1}$) were recorded during the monsoon and winter period, respectively. The mean highest (0.059 ± 0.02 mg L$^{-1}$) and the lowest (0.009 ± 0.002 mg L$^{-1}$) Cr concentrations were recorded in the monsoon and pre-monsoon periods, respectively. The water samples, on the other hand, had just a total average trace quantity of Cd (0.001 ± 0.0002 mg L$^{-1}$) and no data from the post-monsoon period, respectively.

### 3.2. Heavy Metals Concentration in Fish

#### 3.2.1. Lead (Pb) Concentrations

The concentrations (µg g$^{-1}$) of lead (Pb) were found to be significantly different across different fish species in the study area ($p < 0.05$), as shown in Figure 2a–c, respectively. The highest average Pb concentrations (0.388 ± 0.291 µg g$^{-1}$) were recorded from *Labeo rohita*, followed by *Cirrhinus cirrhosus* (0.334 ± 0.236 µg g$^{-1}$) and *L. calbasu* (0.251 ± 0.117 µg g$^{-1}$), respectively.

Moreover, the highest and the lowest average concentrations of Pb in all species were found to be significant ($p < 0.05$) during the monsoon and winter periods, respectively. Therefore, the decreasing order of Pb in all the three species was followed by the monsoon > pre-monsoon > post-monsoon > winter season, except for the *L. calbasu*, respectively.

#### 3.2.2. Chromium (Cr) Concentrations

The total mean concentrations (µg g$^{-1}$) of Chromium (Cr) in different observed fish species at different sites in the Hakaluki *haor*, as shown in Figure 3a–c, were not significantly different ($p < 0.05$) as well. However, the highest mean concentrations in all species were found to be significant ($p < 0.05$) during the monsoon periods, respectively. Moreover, *L. calbasu* possessed significantly higher concentrations of CrCr (0.217 ± 0.170 µg g$^{-1}$), followed by *L. rohita* (0.188 ± 0.031 µg g$^{-1}$) and *C. cirrhosis* (0.176 ± 0.072 µg g$^{-1}$) during the monsoon periods ($p < 0.05$), respectively. However, the decreasing order of Cr in all species followed by monsoon > post-monsoon > winter > pre-monsoon, except in

*C. cirrhosus* species in which the order found as monsoon > pre-monsoon > post-monsoon > winter, respectively.

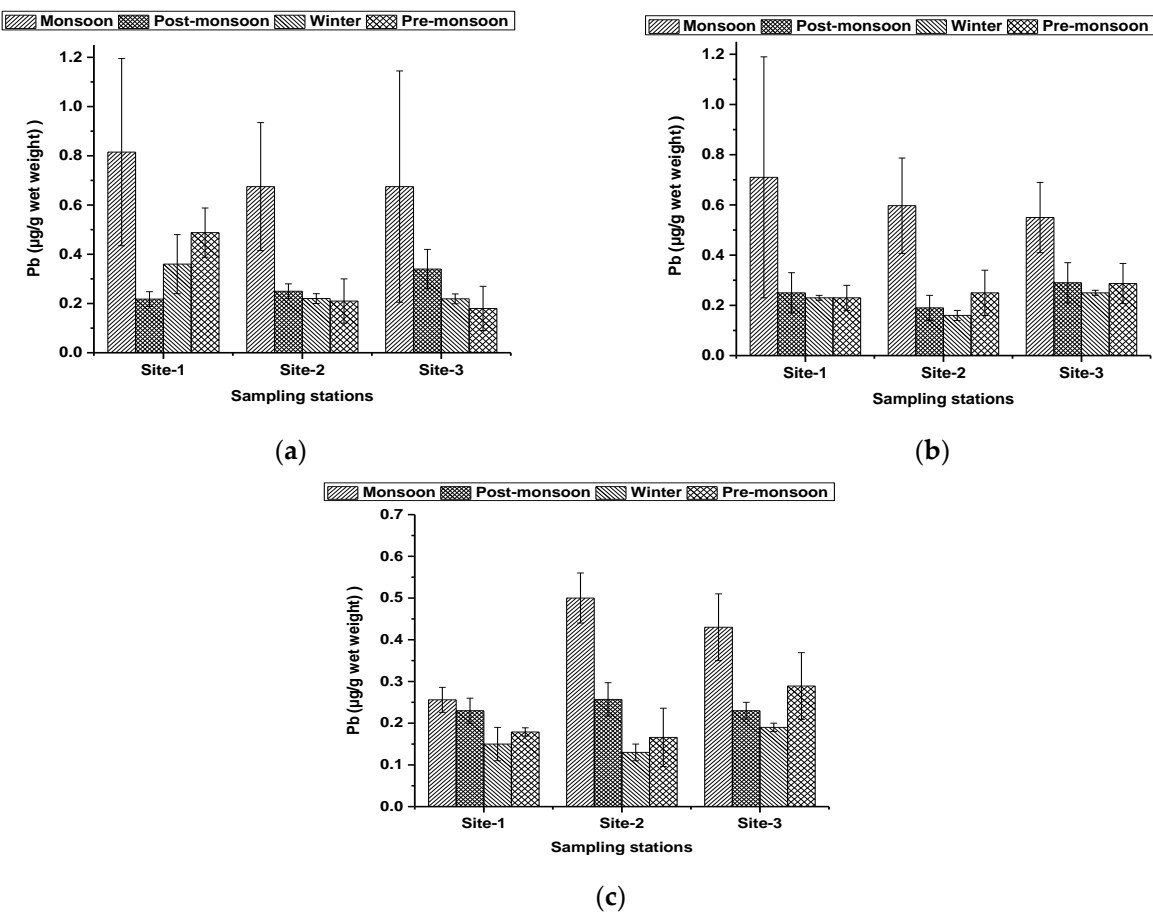

**Figure 2.** (**a**) Concentrations of Lead (Pb) in *L. rohita* fish species; (**b**) Concentrations of Lead (Pb) in *C. cirrhosus* fish species; (**c**) Concentrations of Lead (Pb) in *L. calbasu* fish species.

3.2.3. Cadmium (Cd) concentrations

In practically all seasons, extremely trace amounts of cadmium (Cd) were found in the fish samples from the Hakaluki *haor* at three sampling sites, as shown in Figure 4a–c, respectively. However, the total mean concentrations of Cd were comparatively higher in *L. calbasu* ($0.007 \pm 0.004$ μg g$^{-1}$), followed by *L. rohita* ($0.003 \pm 0.002$ μg g$^{-1}$) and *C. cirrhosis* ($0.001 \pm 0.0001$ μg g$^{-1}$) as well. Moreover, the results of the mean concentration of Cd in three different sampling sites revealed that the highest concentration was observed in the post-monsoon period for *L. calbasu* ($0.009 \pm 0.002$ μg g$^{-1}$) and *C. cirrhosis* ($0.001 \pm 0.0001$ μg g$^{-1}$), though it was recorded for *L. rohita* ($0.005 \pm 0.002$ μg g$^{-1}$) during the pre-monsoon period, respectively. Therefore, during the monsoon season, no Cd data for *C. cirrhossus* ($0.000 \pm 0.000$ μg g$^{-1}$) were recorded.

*3.3. Water Quality Parameters*

Table 4 shows the physico-chemical characteristics of the water column, such as the dissolved oxygen (DO), pH, temperature, and so on. The water temperature values ranged from 18.4 °C to 30 °C between the four seasons, in which the average highest (29.73 °C) and lowest (18.43 °C) values were recorded from the post-monsoon period and winter, respectively. Moreover, the dissolved oxygen (DO) levels were above 5 mg L$^{-1}$ in all seasons; however, the average highest (7.7 mg L$^{-1}$) and lowest (6.56 mg L$^{-1}$) DO also had documented during post-monsoon and winter, respectively (Table 4).

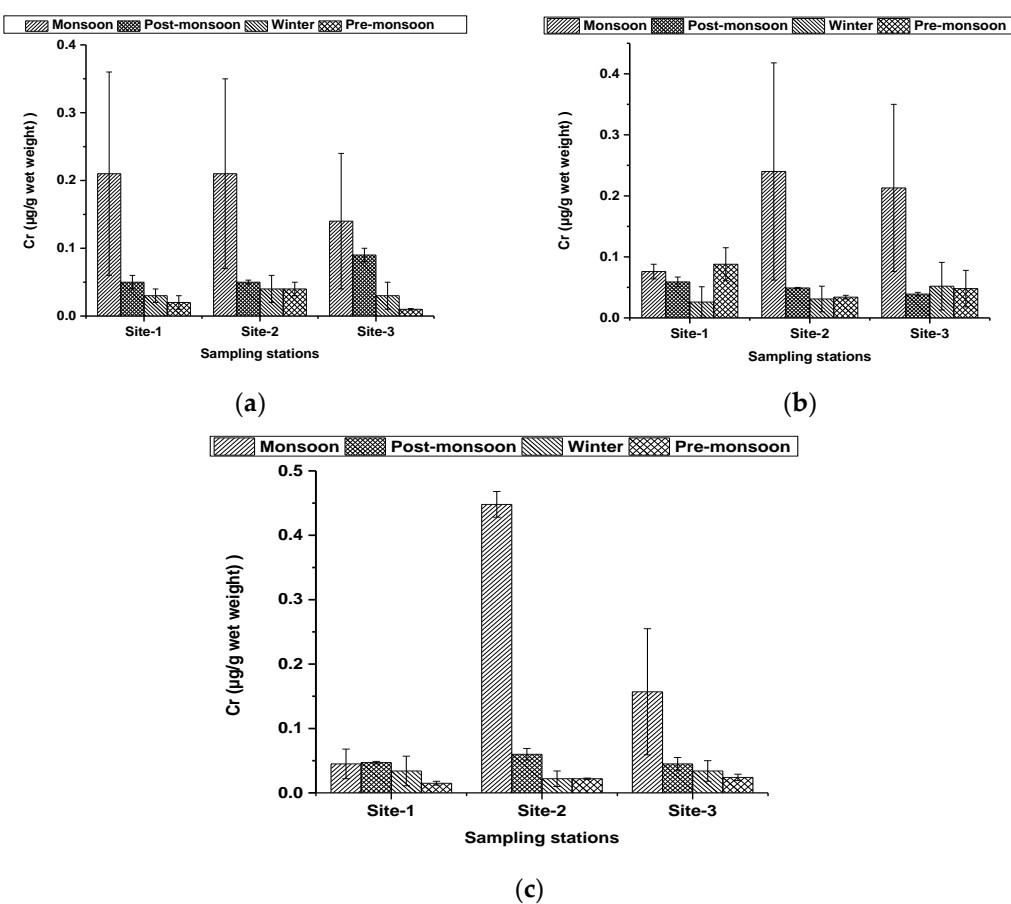

**Figure 3.** (**a**) Concentrations of Chromium (Cr) in *L. rohita* fish species; (**b**) Concentrations of Chromium (Cr) in *C. cirrhosus* fish species; (**c**) Concentrations of Chromium (Cr) in *L. calbasu* fish species.

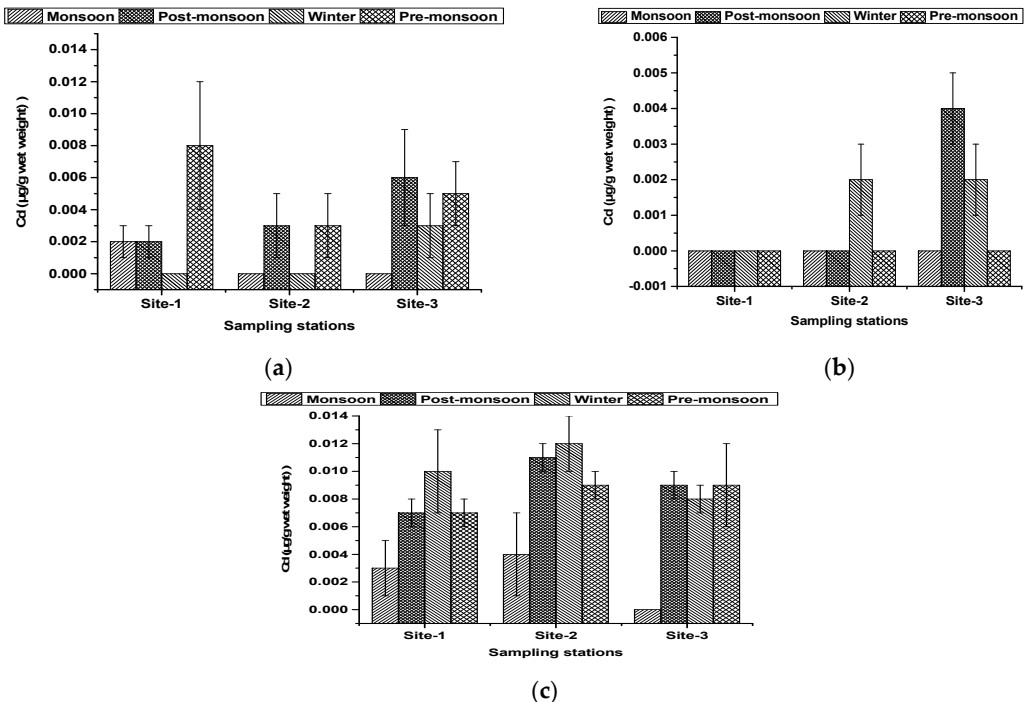

**Figure 4.** (**a**) Concentrations of Cadmium (Cd) in *L. rohita* fish species; (**b**) Concentrations of Cadmium (Cd) in *C. cirrhosus* fish species; (**c**) Concentrations of Cadmium (Cd) in *L. calbasu* fish species.

**Table 4.** Water quality parameters of the Hakaluki *haor* of Bangladesh (Average ± SD).

| Parameters | Seasons | Sites | | | Average ± SD |
|---|---|---|---|---|---|
| | | Site-1 | Site-2 | Site-3 | |
| Air temp. (°C) | Monsoon | 26.4 | 27 | 27.1 | 26.83 ± 0.37 |
| | Post-monsoon | 31.7 | 31.4 | 32 | 31.7 ± 0.3 |
| | Winter | 22 | 22.5 | 22.2 | 22.23 ± 0.25 |
| | Pre-monsoon | 23.7 | 24.9 | 24.6 | 24.4 ± 0.62 |
| Water temp. (°C) | Monsoon | 24.6 | 25 | 24.6 | 24.73 ± 0.23 |
| | Post-monsoon | 29.5 | 29.7 | 30 | 29.73 ± 0.25 |
| | Winter | 18.4 | 18.5 | 18.4 | 18.43 ± 0.06 |
| | Pre-monsoon | 21.5 | 21.7 | 21.7 | 21.63 ± 0.12 |
| DO (mg L$^{-1}$) | Monsoon | 6.3 | 6.7 | 7.1 | 6.7 ± 0.4 |
| | Post-monsoon | 8 | 7.9 | 7.2 | 7.7 ± 0.44 |
| | Winter | 5.6 | 8.1 | 6 | 6.56 ± 1.34 |
| | Pre-monsoon | 7.6 | 6.9 | 7.7 | 7.4 ± 0.44 |
| CO$_2$ (mg L$^{-1}$) | Monsoon | 46.6 | 48.3 | 46.6 | 47.16 ± 0.98 |
| | Post-monsoon | 43.3 | 48.3 | 48.3 | 46.63 ± 2.88 |
| | Winter | 51.6 | 51.6 | 48.3 | 50.5 ± 1.91 |
| | Pre-monsoon | 45 | 41.6 | 50 | 45.53 ± 4.23 |
| pH | Monsoon | 8.1 | 7.9 | 7.7 | 7.9 ± 0.2 |
| | Post-monsoon | 7.7 | 8.2 | 7.9 | 7.93 ± 0.25 |
| | Winter | 7.5 | 7.2 | 7.2 | 7.3 ± 0.17 |
| | Pre-monsoon | 7.3 | 7.8 | 7.9 | 7.66 ± 0.32 |
| Hardness (mg L$^{-1}$) | Monsoon | 105.45 | 114 | 113.05 | 110.83 ± 4.69 |
| | Post-monsoon | 74.96 | 105.45 | 114 | 98.13 ± 20.52 |
| | Winter | 114 | 108.3 | 94.05 | 105.45 ± 10.28 |
| | Pre-monsoon | 101.65 | 111.26 | 113.16 | 108.69 ± 6.17 |
| Ammonia (mg L$^{-1}$) | Monsoon | 0.43 | 0.16 | 0.23 | 0.27 ± 0.14 |
| | Post-monsoon | 0.5 | 0.16 | 0.06 | 0.24 ± 0.23 |
| | Winter | 0.33 | 0.53 | 0.06 | 0.31 ± 0.24 |
| | Pre-monsoon | 0.43 | 0.43 | 0.13 | 0.33 ± 0.17 |

## 4. Discussion

### 4.1. Heavy Metals Concentration in Water

The heavy metal concentrations in various fish species and water are also influenced by variations in aquatic ecosystems with regards to the kind and amount of water pollution, the chemical form of metals in the water, water temperature, pH value, dissolved oxygen concentration, and water clarity [24]. Therefore, the total mean concentration of Pb (0.125 ± 0.058 mg L$^{-1}$) in all seasons and sites of the present study was found to be higher than the permitted limit of 0.05 mg L$^{-1}$ set by WHO [25]. A similar study was also conducted on the Daleshwari River and Turag River in Bangladesh, in which the Pb concentrations found ranged from 0.04 to 0.05 mg L$^{-1}$ and 0.10 to 0.63 mg L$^{-1}$, respectively [26,27], which is consistent with the current study's findings. However, another study documented a maximum Pb concentration of 13.12 ± 0.18 mg L$^{-1}$ from the Argungu River water sample [28], which is higher than the findings of the present study.

Furthermore, the current findings of the total mean concentration of Cr (0.026 ± 0.012 mg L$^{-1}$) were recorded to be lower than the recommended limit (0.05 mg L$^{-1}$) set by the WHO (2008). However, the levels of Cr surpassed the recommended limit in site-1 (0.083 ± 0.040 mg L$^{-1}$) and site-2 (0.059 ± 0.032 mg L$^{-1}$), respectively, during the monsoon period. Different studies discovered a mean Cr concentration of 0.0346 mg L$^{-1}$ in Meghna River water and a mean Cr concentration of 0.021 mg L$^{-1}$ in the Lerma River of Mexico [29,30], which are both similar to our study findings. However, Ali et al. [23] discovered a mean Cr concentration of 0.069 mg L$^{-1}$ in the Karnaphuli River water, while another study revealed a mean Cr concentration of 0.587 mg L$^{-1}$ in Buriganga River

water [31], both of which were higher than the current investigation and both exceeded the WHO [25] recommended limit, respectively. Thus, the heavy metal concentrations differed from station to station, which could be attributed to river water flow, industry locations, municipal and commercial drainage systems, agricultural runoff and so on [29].

Moreover, cadmium (Cd) is a very poisonous metal that humans can be exposed to by inhalation and ingestion, resulting in acute and chronic intoxications [32]. However, the quantity of cadmium in the water samples from the Hakaluki *haor* was determined to be less than the EU [33] standard of 0.05 mg L$^{-1}$ and the WHO standard of 0.005 mg L$^{-1}$ (2008). This indicates that the water in Hakaluki haor is Cd-free. Similar studies were conducted by Ali et al. [23] and Ahmed et al. [26], respectively, in the Karnaphuli River and Daleshwari River water of Bangladesh, who found a mean Cd concentration of 0.006 mg L$^{-1}$, which is consistent with the present study's findings as well.

*4.2. Heavy Metals Concentration in Fish*

Fish with high amounts of critical and harmful metals have been demonstrated to have a significant effect on the ecosystem, posing a risk to the ecosystem's inhabitants [34]. However, chronic infections in people can be caused by excessive accumulation of toxic metals through interactions with the food chain and even at low concentrations over the course of a lifetime, leading to a variety of toxic and non-toxic health issues as well [35]. Therefore, in the present study, the mean concentrations of Pb in all seasons were found to be below the maximum permissible limits (0.3 µg g$^{-1}$) of heavy metals in fish muscle set by FIQC [36], except in monsoon periods, respectively. Though the total average Pb concentrations were comparatively higher in *L. rohita* (0.388 ± 0.291 µg g$^{-1}$), therefore, the study led by Islam et al. [17] in the *haor* regions of Bangladesh found the highest Pb during the pre-monsoon period, which is different from the present study's findings. It could be attributable to the fact that the concentrations of heavy metals in the water were substantially higher in the wet season than in the other seasons (Table 3), resulting in the fish ingesting more heavy metals during the monsoon period. Additionally, a different study found that Pb, Cd, Cu, Mn, and Zn concentrations in fish from the Bangshi River were 0.21, 0.02, 0.65, 3.45, and 5.81 mg kg$^{-1}$, respectively, during the dry season [37]. The concept is the same as the present findings.

Furthermore, the mean concentrations of Cr observed in this study were below the maximum permissible limit (1.0 µg g$^{-1}$) of heavy metals in fish muscles set by the FAO [38] and FIQC (2014). Although an organism can ingest metals directly or through food particles from a water body [26], there was less Cr in the water (Table 3), supporting the current findings of reduced Cr as well. However, another study concerning *Glossogobius giuris* from the River Shitalakhya in Bangladesh [26] discovered that the concentration of Cr (6.92–12.23 µg g$^{-1}$) on a dry weight basis was greater than the current study's findings. These differences in metal concentration in the same species might be found depending on geographical location, capture season, and so on [7,39].

Similarly, the mean concentrations of Cd in the present findings were found to be lower than the maximum permissible limit (0.05 µg g$^{-1}$) according to FAO (1983) and FIQC (2014), respectively. This is because the water of Hakaluki *haor* contains very low amounts of Cd (Table 3). This result indicates the good water quality as well as the comparatively healthy fish species that can be consumed from the largest *haor* in Bangladesh. However, some studies have also found comparatively higher concentrations of Cd (0.09–0.87 µg g$^{-1}$ in dry weight) in some edible fishes from the Bangshi River of Bangladesh [40].

Finally, it can be summarized that, because of differences in numerous aspects such as aquatic conditions, nutrition, and the kind and quantity of water pollution, it is extremely difficult to compare metal concentrations even between the same tissue in different fish. Many experts think that heavy metal bioaccumulation in fish is species-dependent. Therefore, the size (body weight and length), gender, age, and growth rates of fish species, as well as the types of tissues analyzed and physiological circumstances, can all be connected to variations in heavy metal concentrations in different fish species [41,42].

### 4.3. Water Quality Parameters

The physico-chemical parameters are crucial since they have a big impact on water quality [23,43]. Furthermore, the decrease in water quality has an impact on aquatic life. Among the extrinsic elements that influence aquatic ecology, temperature is one of the most important elements. However, the average water temperature was found to be between 25 and 30 °C in the present study, which was within the WHO's permitted limits [25]. Additionally, the average DO was found to be lowest ($6.56 \pm 1.34$ mg L$^{-1}$) during winter and highest during the post-monsoon ($7.7 \pm 0.44$ mg L$^{-1}$) period, respectively. This could be due to the lowest mixing of air into water as well as less rainfall during the winter [23]. Additionally, the quality of water is usually described using the term "hardness". The average hardness value ($110.83 \pm 4.69$ mg L$^{-1}$) in this study was found to be highest during the monsoon season, and the lowest value ($98.13 \pm 20.52$ mg L$^{-1}$) was recorded in the post-monsoon, respectively. This could be the result of higher levels of contaminants being washed away by heavy rain and mixed with *haor* water during the monsoon than in other seasons [21,44]. Moreover, the average ammonia concentrations ($0.24 \pm 0.23$ to $0.33 \pm 0.17$ mg L$^{-1}$) were found within the decreased range of the toxicity levels set by the Inland Fisheries Advisory Commission (1973) [44]. Therefore, the water quality of the Hakaluki *haor* was found to be less polluted but will be a matter of concern in the near future if the appropriate authority does not take necessary action.

### 5. Conclusions

According to the findings of the study, there were substantial differences in the seasonal bioaccumulation of selected heavy metals from Hakaluki *haor* in the water and among fish species. Metals in water and fish species may vary seasonally due to physicochemical and biotic variables in the *haor*, which influence metal bioavailability. However, Pb, Cr, and Cd concentrations were most likely below the maximum permissible limits set by EC, FAO, and FIQC, though Pb was comparatively higher during the monsoon period. Therefore, the present study suggests that to ensure food safety and safeguard public health from potential hazards, proper monitoring and management action should be performed to limit the metal contamination of Hakaluki *haor*, especially in the monsoon season.

**Author Contributions:** A.S.M.S.R.B.: Design, formulation, field data collection, and draft writing of the manuscript. S.K.B.: Data formulation, editing, review, and writing of the manuscript. M.M.H. and M.M.H.K.: Design, editing and review of the manuscript. K.K.M.: Review and writing of the manuscript. S.K.M.: Data analysis, editing and review of the manuscript. All authors have read and agreed to the published version of the manuscript.

**Funding:** This research received no external funding.

**Institutional Review Board Statement:** Not applicable.

**Informed Consent Statement:** Not applicable.

**Data Availability Statement:** The datasets used and analyzed during the current study will be provided on request to the corresponding author.

**Conflicts of Interest:** The authors declare they have no conflict of interest.

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
