# Peer review of "Seasonal Dynamics of Heavy Metal Concentrations in Water and Fish from Hakaluki Haor of Bangladesh"

_conservation, doi:10.3390/conservation2030032_

Round 1

Reviewer 1 Report

Dear authors,

There are some points where the paper should be improved. Please consider the following suggestions.

1.     A presentation of similar research is missing. I suggest to include relevant references and their results in the introduction. This would help the reader to evaluate the significance of your work and understand its contribution.

The studies included in the discussion could be considered.

2.     I suggest to improve the positioning and alignment of the tables in the text.

3.     There are some linguistic mistakes and language improvement is possible. I suggest the authors to review English language and style once again.

Author Response

Responses to Reviewer-1 Comments

Title: SEASONAL DYNAMICS OF HEAVY METAL CONCENTRATIONS IN WATER AND FISH FROM HAKALUKI HAOR OF BANGLADESH

Manuscript ID: conservation-1795643

Reviewer Comments

Reviewer 1

Dear authors,

There are some points where the paper should be improved. Please consider the following suggestions.

>>Responses: Thank you for your valuable comments about our work. Based on the comments and suggestions put, the essential amendment has been made where we addressed all the issues. We highlighted all the corrections by track change. Thank you again.

Specific comments in the manuscript

Comments

Responses

A presentation of similar research is missing. I suggest to include relevant references and their results in the introduction. This would help the reader to evaluate the significance of your work and understand its contribution.

The studies included in the discussion could be considered.

The author has added relevant research with reference in the introduction section for a better understanding of the significance of our study. 

I suggest to improve the positioning and alignment of the tables in the text.

The author has improved the positioning and alignment of the tables in the revised manuscript.

There are some linguistic mistakes and language improvement is possible. I suggest the authors to review English language and style once again.

The author has reviewed the English language and style once again according to suggestions and changed using track change option. 

Reviewer 2 Report

Comments to authors

The authors aimed to assess the contamination of three heavy metals (Pd, Cr, Cd) in the three key fish species and water in the Hakaluki haor of Bangladesh, across seasonal changes. 

To address this, they collected samples of fish tissue and water samples, measured heavy metals through AAS and displayed the results by species and season. This is an active research area and would be interesting to readers. 

I enjoyed reading this article. Your work is valuable and can be better highlighted by (1) a careful review of the grammar and phrasing of the manuscript (2) modifications and suggested additions to the manuscript, especially around the introduction and methods. 

Specific comments:

  1. Please define what a haor is for novice readers.

  2. Please review manuscript. Many areas of the manuscript could be reviewed for phrasing. For example, in the introduction: “that can causing cytotoxic, mutagenic and carcinogenic effects in animals” should be “...can cause cytotoxic…” Please have this manuscript reviewed carefully, as we want to ensure the ideas are clear and concise for readers. 

  3. Why did you choose Pb, Cr, Cd as your key metals? Justify this.

  4. Your sample size of 72 total fish and 24 per species needs to be justified. You can do a power analysis to see if you have the power to detect a certain change in concentration of these metals with the current sample you have. If you find you are underpowered, you would need to add more samples to your study. Figure 2.1 suggests you may be underpowered to detect L. rohita during the monsoon.

  5. In the methods, you clearly state your methods. However, I would like to learn more about if others have used similar methods. Please cite those similar papers to show that these methods are well-vetted. I like your methods overall, but they need to be further justified.

  6. Figure 3.1: The font in the axes labels for most figures are a bit small, can you increase the size a bit?

Author Response

Responses to Reviewer-2 Comments

Title: SEASONAL DYNAMICS OF HEAVY METAL CONCENTRATIONS IN WATER AND FISH FROM HAKALUKI HAOR OF BANGLADESH

Manuscript ID: conservation-1795643

Reviewer Comments

Reviewer 2

The authors aimed to assess the contamination of three heavy metals (Pd, Cr, Cd) in the three key fish species and water in the Hakaluki haor of Bangladesh, across seasonal changes. 

To address this, they collected samples of fish tissue and water samples, measured heavy metals through AAS and displayed the results by species and season. This is an active research area and would be interesting to readers. 

I enjoyed reading this article. Your work is valuable and can be better highlighted by (1) a careful review of the grammar and phrasing of the manuscript (2) modifications and suggested additions to the manuscript, especially around the introduction and methods. 

>>Responses: Thank you for the encouraging comments about our work. We strongly believe that your valuable comments significantly improve the quality of the manuscript. We edited and highlighted by track change wherever we made changes according to your suggestions.

Specific comments in the manuscript

Comments

Responses

Please define what a haor is for novice readers.

The author has defined ‘haor’ in the introduction of revised manuscript.

Please review manuscript. Many areas of the manuscript could be reviewed for phrasing. For example, in the introduction: “that can causing cytotoxic, mutagenic and carcinogenic effects in animals” should be “...can cause cytotoxic…” Please have this manuscript reviewed carefully, as we want to ensure the ideas are clear and concise for readers. 

The manuscript has been reviewed for phrasing in different areas of the revised manuscript for clear and concise ideas for readers.

Why did you choose Pb, Cr, Cd as your key metals? Justify this.

Even in modest amounts, heavy metals' poisonous qualities can have a negative impact on human health. However, some heavy metals, such as cadmium (Cd), chromium (Cr), manganese (Mn), and lead (Pb), are thought to be extremely harmful to humans, causing problems with the liver and kidneys as well as genotoxic carcinogens. That’s why we have chosen to analyse Pb, Cr, Cd as the key metals in our present study.

Your sample size of 72 total fish and 24 per species needs to be justified. You can do a power analysis to see if you have the power to detect a certain change in concentration of these metals with the current sample you have. If you find you are underpowered, you would need to add more samples to your study. Figure 2.1 suggests you may be underpowered to detect L. rohita during the monsoon.

I also do agree with you. Our team is actively working in different water bodies in Bangladesh and I assure you that next time definitely will check doing power analysis to see the appropriateness of the sampling size. Thank you.  

In the methods, you clearly state your methods. However, I would like to learn more about if others have used similar methods. Please cite those similar papers to show that these methods are well-vetted. I like your methods overall, but they need to be further justified.

The author has added a few citations from similar papers to present our methods well-vetted in the revised manuscript.

Figure 3.1: The font in the axes labels for most figures are a bit small, can you increase the size a bit?

The author has checked and edited accordingly.

Reviewer 3 Report

This paper describes about seasonal dynamics of heavy metals in water and fish at the freshwater wetland ecosystem of Bangladesh.

I think this paper can publish after following corrections:

1.     Fig. 1. Font inside figure is too small. Please use large font. Horizontal and vertical caption is too small, cannot read. Please use larger font and clear resolution.

2.     Please make additions to section “Materials and Methods” about the methods of measurement of physical and chemical parameters of water (for example, link: Standard Methods for the Examination of Water and Wastewater, 22nd Edition, Editors: Bridgewater, L., Rice, 491 E.W., Baird, R.B., Eaton, A.D., Clesceri, L.S., APHA-AWWA-WEF: Washington, D.C., 2012; 1496 p., ISBN 492 9780875530130 0875530133).

3.     Title “Materials and Methods”, “Results and title “Discussion should be numbered 2, 3 and 4 respectively.

4.     Table. 3. Mark all heavy metals in bold.

5.     Captions of figures 2.1, 2.2, etc. are made in different font. Please correct.

6.     Section 4.3 Water quality parameters. Please make additions. Discuss the effect of water hardness, the effect of ammonium on the content of heavy metals in water and fish.

Author Response

Responses to Reviewer-3 Comments

Title: SEASONAL DYNAMICS OF HEAVY METAL CONCENTRATIONS IN WATER AND FISH FROM HAKALUKI HAOR OF BANGLADESH

Manuscript ID: conservation-1795643

Reviewer Comments

Reviewer 3

This paper describes about seasonal dynamics of heavy metals in water and fish at the freshwater wetland ecosystem of Bangladesh.

I think this paper can publish after following corrections.

>>Responses: Thank you so much for your valuable comments and suggestions about our work. We strongly believe your valuable corrections/suggestions will improve our manuscript meaningfully. We edited and highlighted all corrections by using track change in our manuscript according to your suggestions.

Specific comments in the manuscript

Comments

Responses

Fig. 1. Font inside figure is too small. Please use large font. Horizontal and vertical caption is too small, cannot read. Please use larger font and clear resolution.

The author has checked and edited the font size.

Please make additions to section “Materials and Methods” about the methods of measurement of physical and chemical parameters of water (for example, link: Standard Methods for the Examination of Water and Wastewater, 22nd Edition, Editors: Bridgewater, L., Rice, 491 E.W., Baird, R.B., Eaton, A.D., Clesceri, L.S., APHA-AWWA-WEF: Washington, D.C., 2012; 1496 p., ISBN 492 9780875530130 0875530133).

The author has added about the determination of physical and chemical parameters of water under the section “Materials and Methods” in the revised manuscript.

Title “Materials and Methods”, “Results” and title “Discussion” should be numbered 2, 3 and 4 respectively.

The author has checked and edited the section numbers in the revised manuscript accordingly. 

Table. 3. Mark all heavy metals in bold.

All heavy metals in the table are marked as bold in the revised manuscript.

Captions of figures 2.1, 2.2, etc. are made in different font. Please correct.

All the captions of figures and fonts are made in the same format accordingly.

Section 4.3 “Water quality parameters”. Please make additions. Discuss the effect of water hardness, the effect of ammonium on the content of heavy metals in water and fish.

The author has revised and added discussion considering hardness and ammonia under section 4.3 “Water quality parameters” in the revised manuscript.
